# Acceptability of a COVID-19 Vaccine among the Saudi Population

**DOI:** 10.3390/vaccines9030226

**Published:** 2021-03-05

**Authors:** Eman Ibrahim Alfageeh, Noor Alshareef, Khadijah Angawi, Fahad Alhazmi, Gowokani Chijere Chirwa

**Affiliations:** 1Primary Health Care Centers Affairs Administration, East Jeddah General Hospital, Jeddah 22253, Saudi Arabia; Emanfageeh@gmail.com; 2Department of Health Services and Hospital Administration, Faculty of Economics and Administration, King Abdulaziz University, Jeddah 80200, Saudi Arabia; kkangawi@kau.edu.sa (K.A.); fnalhazmi@kau.edu.sa (F.A.); 3Economics Department, Chancellor College, University of Malawi, Zomba, Malawi; gowokani@gmail.com

**Keywords:** COVID-19, hesitancy, public, intention, Saudi Arabia, vaccine

## Abstract

To investigate the associated factors underlying vaccination intentions for Coronavirus Disease 2019 (COVID-19), an online cross-sectional survey was conducted among adults 18 years or over in the Kingdom of Saudi Arabia. Data were collected between 8 and 14 December 2020. A logistic regression analysis was employed to examine and identify the variables associated with vaccination intentions for COVID-19, with the odds ratio (OR) and 95% confidence interval (CI) also calculated. A total of 2137 respondents completed the questionnaire. Overall, about 48% of Saudi adults were willing to receive the COVID-19 vaccine. Participants had stronger intentions to receive a vaccination if they resided in the southern region (OR: 1.95; 95% CI: 1.21–3.14), received the seasonal influenza vaccination in the past (OR: 1.52; 95% CI: 1.17–1.97), believed in mandatory COVID-19 vaccination (OR: 45.07; 95% CI: 31.91–63.65), or reported high levels of concern about contracting COVID-19 (OR: 1.91; 95% CI: 1.29–2.81). Participants were less likely to have an intention to be vaccinated if they had a history of vaccine refusal (OR: 0.28; 95% CI: 0.19–0.40). The low acceptance rate among the Saudi population should be targeted with multifaceted interventions aimed at raising awareness and emphasizing the safety and efficacy of the COVID-19 vaccine.

## 1. Introduction

The outbreak of Coronavirus Disease 2019 (COVID-19) has led to the tragic loss of many human lives, as well as the imposition of enormous economic and social disruption across the world [1,2]. Along with protective measures, such as social distancing and quarantine, an efficacious vaccine will be the best strategy for mitigating the spread of COVD-19 and promoting positive clinical and socioeconomic outcomes [3].

Vaccines represent one of the top public health achievements in the 21st century, and the use of immunization has been a major contributor against preventable diseases. However, vaccine hesitancy is a growing challenge for immunization [3]. Indeed, the World Health Organization (WHO) documented vaccine hesitancy as one of the 10 major threats to global health [3]. Vaccine hesitancy is a complex issue and the factors influencing this phenomenon are highly variable across populations [4,5,6]. Several studies found that vaccine hesitancy is linked to certain factors, such as the side effects of the vaccine, misconceptions about the need for vaccination, vaccination history, lack of confidence in the health system, a lack of vaccine and disease literacy, the severity of the disease, the safety and efficacy of the vaccine, and whether the vaccine is provided for free by the government [4,5,6].

Worldwide, countries are putting great effort into the development of a vaccine against the COVID-19 virus [7,8,9]. The success of immunization against COVID-19 depends on adequate vaccine coverage and high vaccine uptake rates among the population. Recent studies have documented uncertainties and skepticism regarding the COVID-19 vaccine as a result of the public’s mistrust of authorities and misinformation spread via social media [10,11,12]. Furthermore, similar fluctuations in COVID-19 vaccination intentions in Australia and France have been reported [11,13].

Since the beginning of the pandemic, the Saudi government has taken a proactive stance in mitigating the spread of the virus; however, these efforts on their own are not effective, and the speedy rollout of a COVID-19 vaccine remains a critical component of any country’s strategy in putting an end to the pandemic [14]. The results of national studies suggest that uncertainty and unwillingness with regard to vaccination against COVID-19 pose challenges in achieving the vaccination coverage needed to reach herd immunity [15]. Thus, there is a need to understand the factors associated with vaccination willingness in Saudi Arabia given that it has a publicly funded healthcare system and coronavirus-related testing and treatment have been provided to both national and non-national residents at no cost. Although a previous study assessing the acceptance of a hypothetical COVID-19 vaccine was conducted among the general public of the Kingdom of Saudi Arabia (KSA) [16], the current study sheds more light on vaccine acceptance by further investigating determinants, such as past vaccination behavior, health status, and support for compulsory vaccination. Following the recent approval of the Pfizer–BioNTech COVID-19 vaccine (Pfizer Inc. and BioNTech SE) in Kingdom of Saudi Arabia (KSA) coupled with the spread of false information [17], it is crucial to assess the acceptance of vaccination at this time since vaccination decisions can be multifactorial and can change over time. Therefore, this study aimed to determine acceptance of the COVID-19 vaccine and investigate factors affecting intentions to be vaccinated against COVID-19 among the general public in KSA. By gaining a deeper understanding of the estimates of COVID-19 vaccine acceptance and influencing factors, health authorities and policymakers can develop evidence-informed communication strategies aimed at building confidence in a vaccine developed in record time, thereby improving vaccination uptake among the general population.

## 2. Materials and Methods

### 2.1. Study Design and Sample

A cross-sectional online survey was carried out among adults in the KSA between 8 December 2020 and 14 December 2020. Participants were invited to complete a self-administrated survey using SurveyMonkey Inc (San Mateo, CA, USA). A link to the survey was distributed via different social media platforms, including WhatsApp Inc. (Mountain View, CA, USA) and Twitter (San Francisco, CA, USA). Eligibility criteria included being 18 years or older and being a current resident in the KSA.

A total of 2319 adults aged 18 years or older participated in the study, of which only 2137 completed the survey (92% completion rate). The average time for completing the survey was 4 min. We utilized a simplified snowball sampling technique, where participants were requested to share the online link of the survey with their contacts. Given the current COVID-19 situation, an online approach was deployed. By using the online approach, further physical contact was avoided, as this might pose a risk of spreading COVID-19. The online approach was also used to generate valid samples in similar studies in KSA and other countries [18].

### 2.2. Measures

The survey was developed on the basis of previous studies assessing the acceptance of novel vaccines for emerging infectious diseases [19,20,21,22,23,24]. The survey consisted of three sections. In the first section, participants were asked to report their age, gender, marital status, educational attainment, monthly income, employment status, and region of residence. The second section gathered information on participants’ health status, vaccination history, and perceived risk of COVID-19. Participants were also asked if they had a chronic disease, had previously contracted or currently had COVID-19, and if any of their family members had previously contracted or currently had COVID-19. In the third section, participants were asked about their acceptance of the COVID-19 vaccine. All questions were closed-ended (see Appendix A). The questionnaire was piloted by a group of experts in the field. The pilot respondents offered valuable feedback on the content of the questionnaire, and inappropriate questions were accordingly modified. The data provided via this sample were not included in the final analysis.

### 2.3. Outcome Measure

To measure vaccination acceptance, participants were provided with the following statement: “Scientists around the world are currently working on a vaccine that could prevent people from contracting COVID-19. It is hoped that the vaccine will become available in a few months”. The participants were then asked the following question: “In the case that a COVID-19 vaccine becomes available in the next few months, with an effective rate of the COVID-19 vaccine between 90% and 95%, would you be willing to get the COVID-19 vaccine if it was provided for free by the government?”

### 2.4. Explanatory Variables

Our choice of explanatory variables was informed by previous studies that investigated intentions regarding vaccination against viral infections [4,5,6,12,13]. Sociodemographic characteristics, such as marital status, age, and gender, were controlled for. To account for the spatial distribution of the variables, we also controlled for the region in which the respondent resided. We also took into account the economic status of the household; as such, monthly income was used. The age variable was divided into five categories: 18–29, 30–39, 40–49, 50–59, and ≥60. The reference group was 18–29. Gender was captured as a binary variable, where a value of one represented male and a value of zero represented female. Marital status was captured as a binary variable, where a value of one was used for marriage and a value of zero was used otherwise (including single, widowed, and divorced). Education level was divided into three categories: high school or below (reference category), bachelor’s degree, and postgraduate degree. Employment status was also categorized into six groups: government employee (reference group), nongovernment employee, self-employed, students, retired, and unemployed. Each was assigned a value of one if in that category and a value of zero otherwise. The region status covered all 13 administrative regions in the KSA and was grouped into five categories: central (reference category), west, east, north, and south. Monthly income (1 Saudi riyal (SAR) = 0.27 United States dollars (USD)) was grouped into eight categories: ˂SAR 3000 (reference category), SAR 3000–<SAR 5000, SAR 5000–<SAR 7000, SAR 7000–<SAR 10,000, SAR 10,000–<SAR 15,000, SAR 15,000–<SAR 20,000, SAR 20,000–<SAR 30,000, and ≥SAR 30,000.

We also controlled for having a history of chronic conditions and coded this as a value of one if the respondent indicated a history of chronic conditions and a value of zero otherwise. Additionally, we checked if the respondent ever previously received a flu vaccination. If so, this was assigned a value of one and a value of zero otherwise. All respondents who had contracted COVID-19 in the past were assigned a value of one and a value of zero otherwise. Those who had a family member who had previously contracted COVID-19 were coded with a value of one and a value of zero otherwise. In addition to the above, we assessed the perceived risk of COVID-19 to people in Saudi Arabia. As such, three categories were used: minor risk or no risk, moderate risk, and significant or major risk. Furthermore, we assessed whether people believed that the COVID-19 vaccine should be compulsory for all citizens and residents in Saudi Arabia. Respondents who believed that it should be compulsory were assigned a value of one and a value of zero otherwise.

### 2.5. Statistical Analyses

This study employed univariate, bivariate, and regression analyses. The univariate analysis produced descriptive statistics, which were generated to produce summary tables for the study variables. These helped us to understand the distribution of socioeconomic characteristics in the data [25]. Using a chi-squared test, bivariate analysis was conducted as a cross-tabulation between all variables and the dependent variable of interest. A logistic regression analysis was employed to examine and identify the variables associated with intention to receive the COVID-19 vaccine, with the odds ratio (OR) and 95% confidence interval (CI) also calculated. The choice of logistic regression was informed by econometric theory, as well as empirical research [26,27,28]. To ensure the robustness of the results, we adjusted for age and gender in three different models (see Appendix B): Model 1 includes age and gender only; Model 2 includes all variables except for gender and age; Model 3 includes all variables except for age. However, no significant differences in the results were observed. All analyses were conducted using STATA 15.1 software (StataCorp LP, TX, USA).

### 2.6. Ethical Consideration

Before progressing to the survey, participants were asked to provide an online informed consent. Participants were informed about the study’s aims and objectives, advised of their ability to withdraw from the study at any time without providing a reason, and assured that the information and opinions provided would be anonymous and confidential. All procedures performed in this study involving human participants complied with the institutional and/or national research committee’s ethical standards and the 1964 Helsinki Declaration and subsequent amendments or equivalent ethical standards. This research was reviewed and given a favorable opinion by King Abdulaziz University. The study was designed and conducted in accordance with the ethical principles established by King Abdulaziz University and, therefore, ethical approval was obtained from the Biomedical Ethics Research Committee, Faculty of Medicine, King Abdulaziz University (Ref-628-20).

## 3. Results

Of the 2319 participants who agreed to participate in this study, 2137 (92%) completed the survey. Among those who completed the questionnaire, 1034 (48.4%) stated that they were willing to accept vaccination for COVID-19 if it was provided for free by the government, with the remaining 52% stating that they were not willing to take it. Table 1 presents the participants’ characteristics and the factors influencing vaccination acceptance. More than half of the participants 1227 (57.4%) were male, 1012 (47%) had a university degree, 971 (45%) were employed in the governmental sector, and 1411 (66.0%) had a monthly income of >15,000 SAR (1 SAR = 0.27 USD).

A total of 521 (24.4%) participants had a chronic disease and 1216 (56.9%) previously received a flu vaccination, whereas 419 (19.6%) previously refused a physician-recommended vaccine. A total of 291 (13.6%) participants had previously contracted COVID-19, and 843 (39.4%) had a family member who had previously contracted COVID-19. Only 765 (35.8%) participants believed that the COVID-19 vaccine should be compulsory. In terms of perceived risk, 870 (40.7%) participants perceived a low or very low risk of contracting COVID-19, and 894 (41.8%) perceived that COVID-19 poses a significant or major risk to the people of Saudi Arabia.

As shown in Table 1, it was found that gender, education level, employment status, region of residence, having received the flu vaccination in the past, having previously refused a vaccination recommended by a physician, perceived risk of COVID-19 to people in Saudi Arabia, concerns about contracting COVID-19, and support for compulsory vaccination were all statistically significant at the 1% level.

Table 2 shows the results of the logistic regression analysis regarding the factors that were associated with an intention to be vaccinated against COVID-19. Males were more likely to have an intention to be vaccinated compared to females (OR: 1.51; 95% CI: 1.10–2.06). Participants with an income level ≥30,000 SAR (OR: 1.85; 95% CI: 0.99–3.45) were more likely to have an intention to be vaccinated against COVID-19 compared with those earning <3000 SAR. Participants who resided in the southern region of the country were more willing to receive the vaccine (OR: 1.95; 95% CI: 1.21–3.14). Participants were also more likely to be vaccinated if they previously received the seasonal influenza vaccination (OR: 1.52; 95% CI: 1.17–1.97), had previously contracted or currently had COVID-19 (OR: 1.48; 95% CI: 0.97–2.27), reported fair (OR: 2.06; 95% CI: 1.49–2.86), high, or very high (OR: 1.91; 95% CI: 1.29–2.81) concerns about contracting COVID-19, and supported mandatory COVID-19 vaccination (OR: 45.07; 95% CI: 31.91–63.65). Participants were less likely to have an intention to be vaccinated if they previously refused a physician-recommended vaccination because of doubts surrounding it (OR: 0.28; 95% CI: 0.19–0.40).

## 4. Discussion

Our study aimed to understand the factors associated with vaccination intentions in a country with a publicly funded healthcare system. We found that perceived risk toward oneself, past vaccination behavior, contracting COVID-19, and support for compulsory vaccination were linked with vaccination intentions. Demographic factors associated with an intention to be vaccinated were being male and residing in the southern region of the country.

The results revealed that about 48% of the participants reported an intention to be vaccinated with the assumption that the vaccine would be provided for free by the Saudi government. The acceptance rate is considered low in comparison with other countries, many of which had higher acceptance rates ranging from 90% in China to 55% in Russia [2]. The percentage of those willing to be vaccinated in this study is at odds with an earlier study conducted in KSA prior to the availability of the vaccine that showed that 64% of respondents were willing to be vaccinated when a COVID-19 vaccine became available [16]. While this indicates that the acceptance rate for a hypothetical vaccine was higher before the availability of the vaccine, this is not surprising. Given the perceived severity of COVID-19 when it was first detected, there might have been a greater intention to be vaccinated; however, at this point of the pandemic, with relaxed restrictions and the decline in transmission in the country, the intention to be vaccinated may have eroded among the Saudi public. Such a behavioral phenomenon was also observed in previous pandemics. For instance, evidence surrounding the influenza pandemic showed that, during the early days of the pandemic, worries and concerns were intense; however, as the pandemic progressed, those worries lessened [29,30]. Furthermore, contextual influences such as communication and the media environment were shown to be linked to vaccine hesitancy [31]. At the time when this study was conducted, circulation of misinformation on the internet was widespread, which might have triggered anxieties and concerns regarding the vaccine.

Importantly, this study confirmed the positive role of past behavior in influencing vaccination intentions. Past behavior related to previous vaccination acceptance was also linked to the intention to be vaccinated in this study. While individuals who previously refused any type of vaccine because of doubts about it were less likely to have an intention to be vaccinated against COVID-19, those who received the flu vaccination in the past were more likely to accept COVID-19 vaccination. These results are consistent with other findings from the United Kingdom (UK), which showed that being previously vaccinated against seasonal influenza was associated with an intention to be vaccinated against COVID-19 [19]. Evidence from previous studies suggests that interventions aimed at increasing seasonal influenza vaccination among those who are eligible is an effective strategy in achieving high rates of vaccination against a pandemic threat [7]. For instance, following the announcement of the COVID-19 vaccine, the UK government planned to expand the seasonal influenza vaccination program to mitigate preventable risk during the pandemic and improve uptake of these novel vaccines [14]. The expansion of the seasonal influenza vaccination program is seen as an opportunity to support and encourage influenza vaccine uptake and, most importantly, to alert and educate people about the COVID-19 vaccine [32]

The results also revealed that the perceived risk of contracting COVID-19 was associated with vaccination intention. This is in accordance with another study, which found that risk perception is a significant predictor of vaccination uptake against numerous diseases [7]. This could reflect the high recognition of the importance of vaccines in self-protection against the disease among the Saudi population. Vaccination among individuals who previously had COVID-19 is recommended by the Centers for Disease Control and Prevention [33]. The perception of the threat posed by COVID-19 to oneself does not seem to cease with contracting the virus. Interestingly, we found that individuals who had previously contracted COVID-19 were more likely to be vaccinated. This suggests that an individual’s personal past experience significantly shapes their decision regarding vaccination. Additionally, those who support mandatory COVID-19 vaccination were more likely to have an intention to be vaccinated. This support could reflect that respondents continue to perceive the pandemic as a major threat to the health of the population and the Saudi economy, despite the low number of COVID-19 cases in the country.

We also found that males were more likely to have an intention to be vaccinated than females. Emerging evidence suggests that men, due to sex-based biological differences, a high reluctance to seek care, and a high smoking rate, are dying in higher numbers from COVID-19 than females [34]. Additionally, although confirmed COVID-19 cases were reported to be the highest in the central, western, and eastern regions of the country [35], participants from the southern region of Saudi Arabia had the strongest intentions to be vaccinated.

The findings of this study could inform governments and health authorities in designing vaccination programs that are tailored to different population groups in need of behavioral change to enhance vaccination uptake among the public.

This study’s strengths include the large sample size, featuring participants from the 13 administrative regions in Saudi Arabia, and the examination of a wide range of possible correlates. The demographic characteristics of the respondents were similar among the population studied. The novelty of this study is that it highlights the important predictors that may impact vaccination decisions, which have crucial implications for communication strategies for different vaccinations in the future.

However, it is worthwhile examining the possible limitations of this study. This study involved a cross-sectional survey that was distributed through different social media platforms; due to the pandemic, other distribution means were not plausible. This could have affected the representativeness of the sample. Another limitation of this study was that this survey evaluated vaccine acceptance with the assumption that the vaccine would be provided for free, with the costs covered by the government; hence, acceptance rates might have differed if respondents were required to pay out of their own pockets. Moreover, the use of the snowball sampling technique may have affected the representativeness of the sample and led to biases in the responses. Participants who agreed to participate and sent the survey to other contacts may have similar characteristics as they use the same social networks. However, due to the COVID-19 outbreak, social media platforms were the only viable source for collecting survey data responses.

Additionally, this study conducted quantitative research using statistical methods to investigate the Saudi public’s intentions to be vaccinated. Consequently, this might have provided a narrower view of the public’s real intentions to be vaccinated, as their intent could have been better explored through a qualitative inquiry that sought to obtain individual views and thoughts about the issue. The provision of deeper insights and thoughts into the Saudi public’s intentions to be vaccinated could have helped to shed light on the reasons for nonacceptance, which was impossible to communicate via the positivist stance adopted in this research. A key limitation was the study’s cross-sectional design and the lack of available data on nonrespondents. Another limitation is that this study could not imply causality, given that it did not use causal identification methods.

## 5. Conclusions

Ensuring the speedy delivery of the COVID-19 vaccine to the Saudi public is among the country’s strategies for mitigating the profound impact of the pandemic. Despite this, this study documented a low acceptance rate. While 52% of the Saudi population was uncertain or did not report any intentions to be vaccinated, 48% were willing to receive the vaccine. This is concerning given that the reported intentions to be vaccinated may not necessarily predict actual health behavior [2].

To overcome this challenge, the public must be targeted with multifaceted interventions aimed at raising awareness and emphasizing the safety and efficacy of the vaccine. Efforts should also highlight the importance of vaccines in achieving herd immunity and accelerating a return to normalcy. Policymakers can also play an important role in influencing vaccine uptake and fostering effective public health interventions to achieve positive vaccination outcomes.

## Figures and Tables

**Table 1 vaccines-09-00226-t001:** Frequency distribution and chi-square analysis of Coronavirus Disease 2019 (COVID-19) vaccine acceptance. SAR, Saudi riyal.

Variables	Willingness to Accept a COVID-19 Vaccine			
	Yes 1034 (48%)	No 1103 (52%)	Chi-Square	*p*-Value
**Age**				
18–29	246 (23.8)	205 (18.6)	11.2305	0.024 **
30–39	341 (33)	379 (34.4)		
40–49	233 (22.5)	245 (22.2)		
50–59	120 (11.6)	156 (14.1)		
≥60	94 (9.1)	118 (10.7)		
**Gender**				
Female	372 (36)	538 (48.8)	35.759	0.000 ***
Male	662 (64)	565 (51.2)		
**Marital status**				
Unmarried	300 (29)	282 (25.6)	3.1997	0.074 *
Married	734 (71)	821 (74.4)		
**Education level**				
High school or below	290 (28)	229 (20.8)	19.4839	0.000 ***
Bachelor’s degree	485 (46.9)	527 (47.8)		
Postgraduate degree	259 (25)	347 (31.5)		
**Employment status**				
Government employee	477 (46.1)	494 (44.8)	28.9683	0.000 ***
Nongovernment employee	140 (13.5)	158 (14.3)		
Self-employed	39 (3.8)	52 (4.7)		
Student	134 (13)	75 (6.8)		
Retired	108 (10.4)	143 (13)		
Unemployed	136 (13.2)	181 (16.4)		
**Region of residence**				
Central	129 (12.5)	188 (17)	38.9615	0.000 ***
South	187 (18.1)	104 (9.4)		
East	103 (10)	121(11)		
North	19 (1.8)	16 (1.5)		
West	596 (57.6)	674 (61.1)		
**Monthly income**				
˂3000 SAR	209 (20.2)	196 (17.8)	9.6153	0.211
3000–˂5000 SAR	84 (8.1)	64 (5.8)		
5000–˂7000 SAR	63 (6.1)	70 (6.3)		
7000–˂10,000 SAR	111 (10.7)	142 (12.9)		
10,000–˂15,000 SAR	219 (21.2)	253 (22.9)		
15,000–˂20,000 SAR	152 (14.7)	172 (15.6)		
20,000–˂30,000 SAR	97 (9.4)	110 (10)		
≥30,000 SAR	99 (9.6)	96 (8.7)		
**Has chronic conditions**				
No	764 (73.9)	852 (77.2)	3.2605	0.071 *
Yes	270 (26.1)	251 (22.8)		
**Received flu vaccination in the past**				
No	348 (33.7)	573 (51.9)	72.8286	0.000 ***
Yes	686 (66.3)	530 (48.1)		
**Refused vaccination in the past**				
No	951 (92)	767 (69.5)	170.4225	0.000 ***
Yes	83 (8)	336 (30.5)		
**Contracted COVID-19**				
No	899 (86.9)	947 (85.9)	0.5362	0.464
Yes	135 (13.1)	156 (14.1)		
**Family members contracted COVID-19**				
No	642 (62.1)	652 (59.1)	1.9808	0.15
Yes	392 (37.9)	451 (40.9)		
**Perceived risk of COVID-19 to people in Saudi Arabia**				
Minor risk or no risk	149 (14.4)	283 (25.7)	89.2711	0.000 ***
Moderate risk	350 (33.8)	461 (41.8)		
Significant or major risk	535 (51.7)	359 (32.5)		
**Concerned about contracting COVID-19**				
Low or very low	320 (30.9)	550 (49.9)	84.9969	0.000 ***
Fair	377 (36.5)	330 (29.9)		
High or very high	337 (32.6)	223 (20.2)		
**COVID-19 vaccine should be compulsory for all citizens and residents in Saudi Arabia**				
No	315 (30.5)	1,057.00 (95.8)	992.1562	0.000 ***
Yes	719 (69.5)	46 (4.2)		

*** *p* < 0.01, ** *p* < 0.05, * *p* < 0.1.

**Table 2 vaccines-09-00226-t002:** Logistic regression estimates of factors associated with COVID-19 vaccine acceptance. OR, odds ratio; CI, confidence interval.

Variables	OR	95% CI
**Age**		
18–29	1	
30–39	0.94	0.60–1.46
40–49	1.04	0.63–1.71
50–59	0.87	0.49–1.54
≥60	1.2	0.60–2.39
**Gender**		
Female	1	
Male	1.51 **	1.10–2.06
**Marital status**		
Unmarried	1	
Married	0.95	0.66–1.36
**Education level**		
High school or below	1	
Bachelor’s degree	0.85	0.61–1.18
Postgraduate degree	0.87	0.58–1.29
**Employment status**		
Government employee	1	
Nongovernment employee	0.75	0.48–1.16
Self-employed	0.68	0.34–1.36
Student	1.48	0.74–2.95
Retired	0.83	0.48–1.43
Unemployed	1.29	0.77–2.16
**Region of residence**		
Central	1	
South	1.95 ***	1.21–3.14
East	1.31	0.80–2.13
North	1.65	0.68–4.02
West	1.02	0.71–1.45
**Monthly income**		
<3000 SAR	1	
3000–˂5000 SAR	1.64 *	0.92–2.92
5000–˂7000 SAR	1.34	0.73–2.47
7000–˂10,000 SAR	1.11	0.62–1.97
10,000–˂15,000 SAR	0.94	0.55–1.59
15,000–˂20,000 SAR	1.06	0.60–1.87
20,000–˂30,000 SAR	1.29	0.70–2.40
≥30,000 SAR	1.85 *	0.99–3.45
**Has chronic conditions**		
No	1	
Yes	0.98	0.72–1.34
**Received flu vaccination in the past**		
No	1	
Yes	1.52 ***	1.17–1.97
**Refused vaccination in the past**		
No	1	
Yes	0.28 ***	0.19–0.40
**Contracted COVID-19**		
No	1	
Yes	1.48 *	0.97–2.27
**Family member contracted COVID-19**		
No	1	
Yes	1.1	0.84–1.43
**Perceived risk of COVID-19 to people in Saudi Arabia**		
Minor risk or no risk	1	
Moderate risk	1.04	0.73–1.48
Significant or major risk	1.34	0.92–1.95
**Concerned about contracting COVID-19**		
Low or very low	1	
Fair	2.06 ***	1.49–2.86
High or very high	1.91 ***	1.29–2.81
**COVID-19 vaccine should be compulsory for all citizens and residents in Saudi Arabia**		
No	1	
Yes	45.07 ***	31.91–63.65

*** *p* < 0.01, ** *p* < 0.05, * *p* < 0.1.

## Data Availability

The dataset generated and/or analyzed in this study is not publicly available due to privacy and confidentiality agreements, as well as other restrictions; however, it is available from the corresponding author on reasonable request.

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
