# Peer review of "Acceptability of a COVID-19 Vaccine among the Saudi Population"

_vaccines, 2021, doi:10.3390/vaccines9030226_

Round 1

Reviewer 1 Report

Overall, the she current research is done well and deserves to be considered for publication.  If the current study has been focused on willingness to accept COVID-19 vaccine by Pfizer-BioNTech specifically, then the title, abstract, and conclusions should be specified accordingly. Otherwise, a big portion of literature review is missing, particularly on skepticism towards adenovirus-based COVID-19 vaccines, such as Russian Sputnik V (Logunov et al., Lancet, 2021; doi: 10.1016/S0140-6736(21)00234-8), and Chinese CansinoBio (Zhu et al.,  Lancet 2020, doi: 10.1016/S0140-6736(20)31605-6) launched onto the market already (page 3, paragraph 1):

  • Burki, Lancet Respir. Med. 2020; doi: 10.1016/S2213-2600(20)30402-1
  • Callaway, Nature 2020, doi: 10.1038/d41586-020-02386-2
  • Bucci, et al., Lancet 2020, doi: 10.1016/S0140-6736(20)31960-7
  • Caddy, S. Russian SARS-CoV-2 vaccine. BMJ 2020; doi: 10.1136/bmj.m3270

The introductory part and discussion should be extended.

Author Response

Point 1: Overall, the she current research is done well and deserves to be considered for publication.  If the current study has been focused on willingness to accept COVID-19 vaccine by Pfizer-BioNTech specifically, then the title, abstract, and conclusions should be specified accordingly. Otherwise, a big portion of literature review is missing, particularly on skepticism towards adenovirus-based COVID-19 vaccines, such as Russian Sputnik V (Logunov et al., Lancet, 2021; doi: 10.1016/S0140-6736(21)00234-8), and Chinese CansinoBio (Zhu et al.,  Lancet 2020, doi: 10.1016/S0140-6736(20)31605-6) launched onto the market already (page 3, paragraph 1):

  • Burki, Lancet Respir. Med. 2020; doi: 10.1016/S2213-2600(20)30402-1
  • Callaway, Nature 2020, doi: 10.1038/d41586-020-02386-2
  • Bucci, et al.Lancet 2020, doi: 10.1016/S0140-6736(20)31960-7
  • Caddy, S. Russian SARS-CoV-2 vaccine. BMJ 2020; doi: 10.1136/bmj.m3270

The introductory part and discussion should be extended.

Response 1: Thank you for raising this comment.  However, this study aims to investigate willingness to accept a COVID-19 vaccine and not the acceptance of the Pfizer-BioNTech COVID-19 vaccine in particular. We have provided this as an introductory information and to inform the reader about the case in Saudi  Arabia. We have not refereed to the name of the vaccine (Pfizer-BioNTech COVID-19) during data collection.  In the questionnaire, the following statement was provided to elicit respondent’s willingness to accept the vaccine “ In the case that a COVID-19 vaccine becomes available in the next few months, with an effective rate of the COVID-19 vaccine between 90-95%, would you be willing to get the COVID-19 vaccine if it was provided free by the government?”.

Reviewer 2 Report

This article investigates factors potentially affecting the intention to be vaccinated for COVID-19 among the general population in Saudi Arabia. The overall concept of the study is quite simple and resembles similar investigations that have been flourishing in the recent past and have focused their search on factors associated with vaccine hesitancy in different countries and cultures. Nonetheless, the study project has been carried out correctly under the methodology standpoint, and materials and methods are described precisely. Results are also reported in clear and satisfying way and the discussion part is adequate. The topic is of course very current and contributions on the matter, when appropriately conducted, are always important to put light on factors that may affect the adherence of a population to the immunization programme and thus to plan appropriate and tailored catch-up strategies.

The only thing I would like the authors to clarify is what makes their study different to an earlier study that was conducted also in Saudi Arabia and that it is cited in the Discussion, other than the fact that the first was conducted prior to the availability of the vaccine, and thus what this study adds to previous reports in terms of knowledge of factors associated with vaccine hesitancy in the saudi arabian population.

Author Response

In this review, Sempere et al. review the roles of different S. pneumoniae (Spn) CBPs in immune evasion and discuss their potential choice as vaccine candidates. The work is concise and well written but addressing certain suggestion will be necessary to improve the comprehensiveness of the manuscript.

We have rephrased/reorganized the manuscript in several places. We believe that the revised version clarifies the concerns raised by the reviewer.

Line 22-23: non-encapsulated pneumococci.

We thank the reviewer for the observation, it is changed, see line 23.

Line 38: human opportunistic pathogen/ pathobiont

We agree with the reviewer that the term pathobiont also is associated to a colonizer specimen. The paragraph used in the introduction is from the paper by Loughran et al. 2018 (reference 1). They do not use the term pathobiont in their manuscript and therefore we prefer to keep the original sentence to avoid doing interpretations of a manuscript by other group.

Line 41: the word 'host' is repeated twice.

We thank the reviewer for the observation, it is changed, see line 41

Line 48: bacteremic pneumonia, otitis media

We thank the reviewer for the observation, it is changed, see line 48.

Line 58: The authors do not include the now very well accepted ability of pneumococcus to cause myocardial inflammation and acute cardiac events which is distinct from its ability to cause pericarditis (Shenoy et al 2017. PLOS Pathogens, Brown et al PLOS Pathogens 2014)

We agree with the reviewer, we have included the suggestion as you can see in line 57.

Line 68: This sentence needs to be reframed

We agree with the Reviewer, the sentence has been modified, see line 67.

Line 68-71: ....preventing the recognition of the bacteria by different components of the immune system like..? Providing specific examples and respective references will be helpful.

We agree with the reviewer in that the sentence is confusing and we have decided to delete it and complete with other sentence, see lines 67-74.

Line 71: My understanding has always been that capsule actually hinders tissue adhesion, invasion and colonization by acting as a steric hindrance for the bacteria trying to attach to host surface. This is why pneumococcal biologists think the bacteria exhibits phase variation to modulate capsule levels as per its requirement. This statement hence might be inaccurate and needs to corrected to accurately reflect the current state of knowledge and should include primary references to support the statement.

We agree with the reviewer in that the sentence is incorrect and we have decided to delete the statement.

Line 86-88: Please give primary references.

We agree with the reviewer, we have added two references, see line 93, with new references from Winkelstein et al and Hummell et al.

Line 98: …common target of many choline-binding proteins

We changed the title of the section, see line 102.

Line 257: CBPs

We thank the reviewer for the observation, it is changed, see line 261.

More discussion about the extent to which each of the mentioned CBP is conserved or variable across known Spn isolates would be helpful.

We agree with the reviewer, we have added this information to the manuscript in the following CBPs: LytA, LytB, LytC, PspA and PspC. See paragraphs in lines: 290, 309, 315,.341, 352, 371, 409, 457.

Line 310: Do the authors mean to say incubation of Spn strains with anti LytB antibodies induced long chain?

The LytB is essential for cellular separation at the end of cell division and pneumococcal strains lacking LytB form long chains instead of typical diplococci. Corsini and collegues observed that antibodies to LytB induce chaining, which is compatible with enhanced susceptibility to complement-mediated phagocytosis and impaired virulence. This chain formation in S. pneumoniae via antibody-mediated agglutination has been shown to be relevant to enhance complement-mediated recognition and phagocytosis. See line 335.

The authors mention about non-encapsulated pneumococcus (NESPs) but fail to discuss PspK as a vaccine candidate. It needs inclusion in the review as it is a major cell surface protein in NESPs that encapsulates the NESPs.

We have added a paragraph in the Pathogenesis, explaining the importance of PspK, see line 83. Moreover, we have given Pspk as an example of a vaccine candidate in the “Current strategies in the development of pneumococcal proteins as vaccine candidates” section (see line 215). We have not analysed PspK with more detail as it is not a CBP.

Spn and S. aureus show a mutually exclusive colonization trend where one doesn’t colonize a host while the other is in place (Bogaert et al. Lancet 2004, Regev-Yochay et al. JAMA 2004, Lebon et al. CVI 2011) A big caveat to developing vaccine candidates that will abolish Spn colonization is freeing up of the host niche for other more invasive bacteria like S.aureus, etc. In such cases, would vaccinating the host to CBP proteins relevant in pneumonia and IPD but minimally expressed during colonization be useful? Some discussion about the possible consequences of complete removal of Spn from human populations need to be discussed.

We agree with the reviewer about the lack of discussion of the matter, we have added different paragraphs discussing the implications of a potential protein vaccine with sterilized immunity. See paragraphs in lines 272, 309, 341, 491, 517 and 532.

Reviewer 3 Report

Lines 56-59 are more conclusion instead of aim.

A more detailed description of the questionnaire and items included is needed. How many items in total? Were they close or open questions? other? please add as much information as possible.

The authors stated that the questionnaire was developed based on previous questionnaires. Can you explain what changes have been done?

Did the authors validate the new version of the questionnaire? no info are provided.

Please, for more transparency add the questionnaire as supplementary material.

Lines 132-134 Authors stated that they used econometric theory as well as empirical research to perform the logistic regression. Can authors provide more details? What does this theory say? How it has been used in this work?

Why authors did not provide a bivariate analysis to describe differences in the sample?

Why the logistic regression was not adjusted for at least sex and age?

Did the authors explore reasons for no acceptance of the vaccine? According to the 3 C Model proposed by the WHO, vaccine hesitancy is highly dependent on the context. More insight in this perspective can add more knowledge to the field. 

The study has several limitations, only partially covered by the authors. They did not use a validated questionnaire, potentially affecting the validity of collected data and consequently interpretation of them. Statistical analysis was very poor, without any type of adjustment. The snowball sampling highly affect the representativeness of the sample, since people, within their network, share the same behaviors and thoughts. And this is particularly true considering the "Filter Bubble" of the internet and social network as Facebook (used by the Authors). 

Author Response

Dear Reviewer, 

Thank you for your valuable feedback, please see the attachment. 

Round 2

Reviewer 3 Report

Authors improved the quality of their reporting of methods and results. 

Regarding the adjustment, it is well known that a crude model is not enough and many variables can play an important role in driving the association. 

Generally speaking, sex and age are really the basic variables for which an adjusted model should be performed. Moreover, the use of social network, vaccine acceptance, human behaviors, in general, are all sex and age-specific. Furthermore, since it is expected that, due to sampling methods, the sample might be not representative of the whole population, adjusting at least for sex and age might improve the reliability of the results.  Last but not least, all the tools assessing the risk of bias of original research have at least one item related to the statistical method used and specifically, adjustment.

Attaching the questionnaire used (both in English and Arabic) a supplementary material might largely improve the transparency and it can be useful for future researchers interested in the same topic. 

 Extensive editing of English language and style required.

Author Response

Dear Reviewer, 

Thank you for raising these comments. As suggested, we have used the Adjusting method that is in the simplest form and involves estimating various models with and without the adjusting variables.  Our results are stable, and we attached it in the manuscript appendix.  In lines 158-160 we have stated “For results robustness, we have also adjusted for gender and age. However, no changes in the results were observed ( see Appendix A)”. Moreover, extensive English editing was done by the MDPI English editing service. 

Attached is the modified manuscript with track changes.

We appreciate the time and effort you have spent to share your insightful comments. 

Round 3

Reviewer 3 Report

What does "module" mean? (in appendix table) Which differences? Please specify in the main text, presumably in statistical analysis, as well as in the table. 

The full version of the questionnaire should be enclosed to the manuscript.

Author Response

Dear Reviewer, 

Below is a point-by-point response to your comments. 

Point 1: What does "module" mean? (in appendix table) Which differences? Please specify in the main text, presumably in statistical analysis, as well as in the table. 

Response 1: Thank you for raising this is up. Sorry, this was a misspelled word “Module” and is now changed to “Model” , we have also shown the differences between the three models. This has been specified in lines 156-160 “For results robustness, we have adjusted for age and gender in three different models (see Appendix A): Model 1; includes age and gender only, Model 2; includes all variables except for gender and age, Model 3; includes all variables except for age. However, no significant differences in the results were observed” as well as in the table.

Point 2: The full version of the questionnaire should be enclosed to the manuscript.

Response 2: the questionnaire has been enclosed in the Supplementary Materials.

Attached is the edited manuscript with track changes. 

Thank you for your time and effort. 
